# Occupational Deviance Among University Counselors in China: The Negative Predictive Role of Professional Identity and the Moderating Effect of Self-Control

**DOI:** 10.3390/bs15091278

**Published:** 2025-09-18

**Authors:** Tiantian Chen, Xianjun Luan, Shenghong Dong

**Affiliations:** 1School of Education, Jiangxi Normal University, Nanchang 330022, China; 005545@jxnu.edu.cn; 2School of Artificial Intelligence, Jiangxi Normal University, Nanchang 330022, China; 3School of Law, Nanchang University, Nanchang 330022, China; luanxianjun@ncu.edu.cn; 4School of Psychology, Jiangxi Normal University, Nanchang 330022, China

**Keywords:** occupational deviance, professional identity, self-control, moderating effect, university counselors

## Abstract

University counselors in China are anticipated to demonstrate professional conduct as an integral aspect of their vocational responsibilities. Although the existing literature primarily emphasizes normative professional behaviors, there is a notable scarcity of research examining occupational deviance, which is largely attributed to inadequate external regulatory mechanisms. Consequently, a significant gap persists in comprehending the subjective perspectives of counselors within this framework. This study seeks to explore the negative predictive influence of professional identity on occupational deviance among university counselors, as well as to assess the moderating role of self-control in this relationship. In October 2024, a total of 363 full-time undergraduate counselors were recruited using the convenience cluster sampling method. Validated scales assessing occupational deviance, professional identity, and self-control were utilized in this study. Hierarchical regression analysis was performed using the PROCESS macro to test the moderating effect. The mean score for occupational deviance was 2.553 ± 1.230, and the detection rate was 14.05%. A significant negative correlation was observed between professional identity and occupational deviance (r = −0.599, *p* < 0.01). After adjusting for demographic variables, professional identity was identified as a predictor of a decrease in deviance (*β* = −0.477, *p* < 0.001). Furthermore, self-control was found to negatively moderate the relationship between professional identity and occupational deviance (*β* = −0.171, *p* < 0.001), explaining 2.8% of the variance in occupational deviance (Δ*R*^2^ = 0.028). Occupational deviance among counselors is observed to occur at relatively low levels. Professional identity significantly and negatively predicts occupational deviance, while self-control enhances this negative relationship. This study provides novel theoretical perspectives and practical insights intended to standardize the management of occupational behaviors among university counselors.

## 1. Introduction

Within the higher education system in China, university counselors (*hereinafter referred to as counselors*) hold a pivotal role in both ideological and political education and the management of students’ daily affairs, exemplifying distinct characteristics of role multiplicity ([73]). As implementers of national educational policies, counselors fulfill the responsibilities of “educators” by disseminating ideological and political theories through classroom instruction, thematic seminars, and other systematic methods of values cultivation ([22]). Additionally, they assume the administrative functions of “managers,” overseeing student affairs such as awards, grants, disciplinary regulations, and evaluations ([40]). This dual identity extends further into the role of “growth advisors” within the service dimension, addressing students’ practical concerns through individualized counseling and career planning ([14]), while also developing a specialized identity as “psychological guides” in mental health education ([37]). Notably, this complexity of roles does not diminish their educational essence; rather, it enriches the concept of student development through management and service processes. Counselors are required to integrate value guidance into their administrative tasks, embody humanistic care in psychological support, and demonstrate responsibility during crisis interventions ([30]; [25]). It is precisely this quadripartite role matrix consisting of “educator–manager–advisor–guide” that compels counselors to consistently uphold professional ethics transcending singular occupational standards, thereby exerting exemplary value leadership across multiple practical domains ([67]).

Counselors, often designated as the “life mentors” of university students, are inherently tasked with exemplifying professional conduct ([25]). Consequently, existing research on their occupational behaviors predominantly exhibits an exemplar-oriented focus, primarily constructing normative frameworks such as models of professional ethics and dedication. However, critical studies addressing instances of occupational deviance remain notably scarce ([15]; [37]; [40]; [67]; [76]). This structural imbalance in research paradigms has inadvertently reinforced a one-dimensional narrative, often referred to as the “virtue myth,” surrounding the counseling profession, thereby obscuring the complexities inherent within their professional behavioral spectrum ([73]). Drawing on Moral Disengagement Theory ([6]), such deviations may stem from temporary ethical disengagement—wherein individuals justify norm violations as necessary adaptations to systemic contradictions. However, merely exposing this research imbalance and underlying paradox is insufficient to fully explain the deep-seated causes of university counselors’ occupational deviance; a further analysis from the perspective of organizational behavior theory is also required.

From the standpoint of organizational behavior, occupational deviance among counselors—characterized by actions that contravene professional ethical standards and role expectations (e.g., *bias in managing student affairs or neglecting educational duties*)—demonstrates dual pathological features in its underlying mechanisms ([68]). At the institutional supply level, there is a disconnect between goal-setting and implementation feedback in management systems, with institutional blind spots particularly evident in non-quantifiable assessment areas ([14]). At the individual agency level, some counselors experience depletion of professional identity and deconstruction of professional personality, where their internal ethical awareness fails to effectively translate into norm-compliant behaviors ([42]; [73]). This dual collapse of exogenous institutional constraints ([14]) and endogenous individual motivation ([42]; [73]) fosters an environment conducive to the perpetuation of occupational deviance.

### 1.1. Occupational Deviance

Occupational deviance encompasses actions undertaken by personnel within a workplace that transgress established boundaries, thereby violating industry standards, ethical codes, or legal provisions in the course of their duties. Such actions infringe upon the legitimate rights of stakeholders, disrupt institutional operations, and jeopardize public welfare ([33]). The theoretical framework for comprehending occupational deviance is grounded in [50] ([50]) Strain Theory, which asserts that when professionals experience a structural misalignment between their career development aspirations and the available pathways for compliant achievement, occupational deviance may arise as a coping mechanism to mitigate institutional contradictions. In the academic field, some researchers ([36]; [11]) have classified and explored occupational deviant behaviors using the dual criteria of the object affected by the behavior and the nature of the violation. Behavioral targets categorize deviance into organization-related deviance (e.g., *educators fabricating academic records to enhance institutional performance or embezzling public funds*) and interpersonal deviance (e.g., *discriminatory treatment of students from marginalized populations or psychological coercion of colleagues*) ([11]). Violation attributes include ethical deviance and institutional deviance. The former pertains to deviations from implicit industry norms (e.g., educators covertly soliciting material benefits or allocating educational resources based on personal interests), while the latter involves breaches of explicit regulations (e.g., unauthorized profit-driven courses or concealed disciplinary actions) or illegal acts (e.g., manipulating evaluations through financial exchanges or falsifying research data) ([36]). Importantly, this classification system illustrates intersectionality; for instance, when educators accept high-value items from parents under the guise of “academic consultation”, they simultaneously violate integrity clauses in professional ethics (i.e., *ethical deviance*), and if the total value exceeds legal thresholds, commit occupational crimes (i.e., *legal deviance*).

In the analysis of behavioral mechanisms, occupational deviance can be further classified according to its underlying driving factors: economic-interest-driven deviance (e.g., *educators leaking examination questions for personal gain; falsifying invoices to misappropriate funds*) ([36]); emotion-driven deviance (e.g., *declining work quality due to evaluation failures; maliciously altering grading standards following pedagogical disputes)* ([53]); and habitual deviance (e.g., *persistent formulaic teaching; unjustifiably monopolizing students’ autonomous time)* ([11]). This framework effectively distinguishes occupational deviance from occupational burnout. The latter is defined as a reactive state of physical and mental exhaustion resulting from chronic stress, primarily characterized by emotional detachment and reduced work efficacy ([47]). In contrast, occupational deviance represents proactive strategic choices aimed at achieving individual objectives through institutional subversion (e.g., *power transactions*, *process distortion*), fundamentally adhering to a logic of resource optimization rather than mere emotional depletion ([53]). Nevertheless, dynamic interconnections exist: ethical desensitization induced by occupational burnout ([6]) may diminish moral restraint, thereby indirectly facilitating deviance ([7]). Furthermore, certain forms of non-utilitarian occupational deviance (e.g., *sustained low-quality teaching*) may exhibit behavioral patterns akin to burnout due to normative misinterpretation and psychological maladjustment ([36]).

In the context of China’s higher education system, university counselors fulfill a distinctive role by offering ideological guidance and overseeing administrative responsibilities. Nonetheless, these counselors also display specific institutional characteristics that may result in occupational deviance. Such occupational deviance includes actions in student education and management that contravene the professional ethics anticipated of university educators, violate organizational discipline, or negatively impact students. This occupational deviance predominantly manifests in four dimensions: neglect of duties, inequitable rewards and punishments, exploitation of power for personal benefit, and the imposition of demeaning or corporal punishment ([68]).

Such behaviors not only undermine the credibility of educational institutions ([35]) but also exert a significant negative influence on students’ value systems through social modeling effects ([5]). Although empirical studies indicate that the incidence of occupational deviance among counselors remains relatively low ([27]), the potential for such occupational deviance to erode institutional integrity necessitates ongoing vigilance ([36]). Two categories of paradoxes arise in practical contexts: (1) There is an asymmetric ebb-and-flow relationship between professional exemplary behaviors and occupational deviance, wherein conscientious counselors may nonetheless engage in occasional deviant acts (e.g., *emotional corporal punishment*) due to situational pressures ([35]). (2) The phenomenon termed “visibility trap” emerges amid enhanced supervision: improvements in the Ministry of Education’s regulatory system have rendered previously concealed occupational deviance more explicit ([73]).

Existing research predominantly adheres to institutionalist paradigms, emphasizing exogenous variables such as deficiencies in organizational culture and regulatory failures ([45]). However, these researchers frequently overlook the regulatory efficacy of psychological resources from the standpoint of subjective agency ([66]). This scholarly tendency reflects structural limitations within traditional frameworks of professional ethics, which are characterized by an excessive reliance on external heteronomous mechanisms (e.g., *quantitative evaluations and complaint-response systems*) and a neglect of the dynamic adaptive functions of intrinsic factors, such as professional identity and self-regulation. As institutional constraints demonstrate diminishing marginal effects ([35]), adopting a subjective perspective to investigate the generative mechanisms of occupational deviance signifies a cognitive revolution in prevention paradigms. This transition shifts the focus from a reactive approach to a proactive one that fosters agentic self-discipline ([44]). Such a paradigm shift not only addresses theoretical gaps but also offers actionable pathways for the development of a collaborative governance system that integrates both institutional and psychological perspectives ([66]).

### 1.2. Professional Identity and Occupational Deviance

Professional identity, a critical subjective variable in the study of occupational behavior, transcends the conventional understanding of emotional attachment within its theoretical framework. It fundamentally consists of a tripartite integration of professional value cognition, role commitment, and behavioral paradigms ([61]). Specifically, professional identity influences behavioral choices through three dimensions: (1) in the cognitive dimension, it facilitates the transition from awareness of professional norms (*understanding “what ought to be”*) to the internalization of values (*embracing “what must be”*); (2) in the affective dimension, it fosters identity commitment rooted in occupational dignity and a sense of belonging; and (3) in the behavioral dimension, it promotes pro-professional behavioral tendencies that align with occupational ethics ([2]). This tri-dimensional structure not only determines work engagement and organizational loyalty ([51]) but also shapes normative behavioral thresholds through the development of moral schemes ([63]).

A dynamic, bidirectional, and deconstructive relationship—encompassing both positive and negative aspects—exists between professional identity and occupational deviance ([41]). On a positive note, a robust professional identity functions as a protective mechanism against occupational deviance through various psychological processes. From a cognitive perspective, internalized professional norms serve as decision-making anchors, thereby decreasing the likelihood of occupational deviance at its inception ([6]). Affectively, a strong commitment to one’s professional identity induces anxiety through cognitive dissonance when these norms are transgressed ([3]).

Conversely, the negative pathway is one through which we can understand how certain behavioral practices can undermine one’s professional identity ([10]). When counselors repeatedly engage in occupational deviance due to situational pressures (e.g., *performance evaluation metrics*) or resource depletion (e.g., *emotional exhaustion*), they often resort to self-justification strategies ([6]), such as attributing their actions to “systemic flaws” or “necessary choices”, undermining the sanctity of professional norms. The phenomenon of “behavior-identity decoupling” is self-reinforcing: each act of self-justification diminishes the alignment with professional values, ultimately leading to a deterioration of role commitment ([63]). This detrimental cycle underscores that occupational deviance is not merely a consequence of identity deficits but also acts as a catalyst for intensifying identity crises ([63]).

Our focus is on identifying protective factors that may enhance intervention effectiveness. This is particularly in response to current school administration concerns about “how identity development prevents deviant behavior.” We also acknowledge the limitation of cross-sectional designs in rigorously testing bidirectional causality. For these reasons, we prioritize examining professional identity as an antecedent variable predicting occupational deviance. We hypothesize that counselors’ professional identity negatively predicts their occupational deviance.

### 1.3. Self-Control and Occupational Deviance

Among the various individual factors influencing unethical professional behaviors, the role of self-control cannot be overlooked ([73]). Self-control refers to an individual’s ability to autonomously regulate their thoughts, emotions, and behaviors to achieve predetermined goals or adhere to specific standards. It exhibits certain trait-like characteristics while possessing significant acquired plasticity, involving behavioral patterns such as inhibiting impulses, overcoming short-term temptations, and following long-term goal-oriented tendencies ([19]). In the context of educational practice, this regulatory efficacy is manifested in two dimensions of norm maintenance: first, emotional regulation, which serves to prevent the generalization of bias and thereby ensures objectivity in educational evaluations (e.g., *avoiding differential treatment based on personal preferences*) ([1]); second, the activation of cognitive reappraisal mechanisms under pressure, which allows for the transformation of stress responses into compliant decisions (e.g., *maintaining professional composure during conflicts with students*) ([58]). Empirical studies ([27]; [73]) reveal a significant negative correlation between levels of self-control and instances of occupational deviance among counselors, indicating that counselors with higher self-control tend to exhibit fewer occurrences of occupational deviance, and, conversely, those with lower self-control demonstrate a higher propensity for such behavior.

From an evolutionary standpoint, human behavior is often characterized by a tendency to prioritize the maximization of self-interest, which suggests that selfishness may be an instinctual trait ([13]). Nevertheless, in order to sustain social harmony, organizational stability, and personal reputation, individuals are compelled to transcend these innate impulses and conform to established social norms ([55]). This phenomenon indicates that individuals frequently face behavioral dilemmas prior to engaging in occupational deviance ([24]). In such dilemmas, the ability to resist violations of social norms necessitates a greater allocation of self-control resources ([23]). [8] ([8]) assert in their limited self-control strength model that self-control resources are depleted with use, thereby affecting subsequent behavioral outcomes. Individuals who exhibit strong self-control possess sufficient resources to effectively regulate their emotions and actions, thus avoiding occupational deviance ([46]). Conversely, a deficiency in self-control resources not only hinders prosocial and norm-compliant behaviors but also amplifies impulsive tendencies aimed at fulfilling short-term desires, thereby increasing the likelihood of engaging in occupational deviance ([4]; [62]; [69]).

### 1.4. Moderating Effect of Self-Control

The Conservation of Resources Theory ([28]) posits that individuals endeavor to preserve, protect, and acquire resources, which can be categorized into material, conditional, personality-based, and energetic domains, including psychological resources. When psychological resources are limited, depleted, or threatened, they can lead to stress and tension, thereby influencing behavior and attitudes ([54]). In professional settings, counselors face numerous stressors—such as administrative responsibilities, complex student management, and pressures associated with teaching and research—that can deplete their psychological resources ([16]). Professional identity, which serves as a cognitive resource for counselors, becomes particularly vulnerable to depletion under stress, potentially wavering in response to adversity ([73]). This reduction in psychological resources can hinder adherence to professional norms and increase the likelihood of occupational deviance. Self-control acts as a regulatory mechanism that allows individuals to manage and allocate their limited psychological resources effectively ([31]). Counselors who experience a depletion of professional identity resources may find that those with high self-control are better equipped to self-regulate, reallocating sufficient resources to sustain their professional identity and mitigate the risk of occupational deviance ([73]). Conversely, counselors with low self-control may struggle to inhibit undesirable behaviors during periods of resource depletion, thereby diminishing the protective effect of professional identity against occupational deviance and increasing its occurrence ([73]). This highlights the significance of considering the moderating role of self-control when examining the relationship between professional identity and occupational deviance.

Therefore, this study hypothesizes that self-control has a significant moderating effect on the relationship between university counselors’ professional identity and occupational deviance. Specifically, it is proposed that self-control strengthens the negative predictive relationship between professional identity and occupational deviance.

### 1.5. The Goal of This Study

This study investigates occupational deviance among counselors, focusing on the interactive mechanisms between professional identity and self-control. The theoretical significance of this research arises from two notable limitations in existing explanations of occupational deviance. (1) There is a lack of clarity regarding the operational pathways of professional identity. While Social Cognitive Theory ([6]) elucidates that professional identity influences behavioral decisions through the construction of self-concept, current research ([66]; [73]) has predominantly concentrated on verifying correlations between variables, thereby failing to reveal the intrinsic dynamic transmission mechanisms. (2) The moderating mechanisms of self-control remain unclear. The Conservation of Resources Theory ([28]) suggests that the allocation of psychological resources may impact the constraining efficacy of professional identity. However, the existing literature tends to report only simple additive effects, neglecting the boundary conditions that elucidate “how” moderation occurs.

Therefore, this study employs empirical research to uncover the synergistic mechanisms by which professional identity and self-control collaboratively mitigate occupational deviance, surpassing the cognitive limitations of traditional single-variable explanatory models. This research not only enhances the literature on professional ethics but also offers universities empirical evidence to establish psychological screening criteria for assessing professional identity in personnel selection, integrate self-control training modules into professional development programs, and ultimately transition governance models from reactive punishment to proactive prevention.

Specific hypotheses are formulated as follows:

**H1.** *Counselors’ professional identity significantly and negatively predicts occupational deviance*.

**H2.** *Self-control has a significant moderating effect on the negative relationship between professional identity and occupational deviance, enhancing the negative predictive relationship between these variables*.

## 2. Methods

### 2.1. Research Design

This study utilized a cross-sectional survey design employing convenience-based cluster sampling. In September 2024, full-time university counselors from 16 undergraduate institutions across five major cities in East and South China—Nanchang, Changsha, Guangzhou, Hangzhou, and Hefei—were recruited. This research implemented strict inclusion criteria: (1) participants must hold full-time counselor positions at the sampled universities, excluding part-time counselors and administrative staff who also serve as counselors, thereby ensuring the purity of occupational roles; and (2) participants must have at least one year of service, ensuring they have completed their occupational role adaptation and established stable professional behavioral patterns.

Participant recruitment strictly adhered to the principles of voluntary participation and informed consent. When accessing the questionnaire through an online platform, participants first encountered an electronically administered informed consent page. This page outlined this study’s purpose, estimated duration, potential risks and benefits (e.g., *contributing to academic knowledge*), data confidentiality measures (e.g., *anonymization*), and explicitly stated the participants’ right to withdraw unconditionally at any time without justification. The participants were required to read the consent form thoroughly before checking the option “I have read and understood the above information and voluntarily agree to participate in this study” as their electronic signature. Individuals who did not consent were automatically redirected to an exit page, preventing access to the questionnaire. Researcher contact information was provided for any inquiries. Upon completing the consent process, the system automatically advanced to the survey.

Data collection was conducted using Wenjuanxing (https://www.wjx.cn), chosen for its user-friendly interface, and integrated features for questionnaire design, dissemination, and analysis, as well as its widespread adoption for research data collection in China. With the support of student affairs departments at partner universities, the questionnaire was distributed to target participants via unique QR codes. Coordinating staff received standardized training in operational procedures and research ethics, which included guidance on avoiding coercive language and explicitly informing participants of their unconditional right to withdraw. Responses were anonymized, with no institutional or individual identifiers collected, and IP address uniqueness verification was activated to minimize duplicate submissions. Participation was voluntary, fully respecting the autonomy of the participants.

The minimum sample size was determined based on the requirements of exploratory factor analysis ([71]). It was calculated as 5 to 10 times the number of questionnaire items (i.e., *k* = 38; *n =* 190–380). After accounting for invalid response rates, the final minimum target was set at 420. This study initially collected 403 responses. Following a rigorous triple-layer quality control process—excluding submissions with a completion time less than the mean minus three standard deviations (i.e., *less than 62 s*), incomplete responses, and patterned responses (i.e., *where 80% or more of the item answers were homogeneous*)—a total of 363 valid samples were included, resulting in an effective response rate of 90.07%.

### 2.2. Demographic Characteristics of Participants

Demographic characteristics of the valid sample (*n* = 363) are presented in detail in Table 1.

### 2.3. Survey Tools

#### 2.3.1. General Information Questionnaire

A self-designed questionnaire was employed to gather demographic data from participants, encompassing variables such as gender, age, professional title, years of service, marital status, and student caseload. Existing research ([15]; [40]; [73]) indicates that these demographic characteristics may systematically influence professional identity development, the availability of self-control resources, and occupational behavioral standards. Consequently, these variables were treated as control variables in the analysis.

#### 2.3.2. Professional Identity Questionnaire

The University Counselor Professional Identity Questionnaire, created by [41] ([41]), was employed to evaluate the participants’ levels of professional identity. This 20-item instrument encompasses three dimensions: professional cognition (*10 items*), professional volition (*5 items*), and professional behavior (*5 items*). Responses are measured on a 5-point Likert scale, with higher scores reflecting a more robust professional identity. The questionnaire has demonstrated strong psychometric properties across multiple studies (*Cronbach’s α* = 0.87–0.91) ([72]; [75]). In this investigation, it achieved an internal consistency coefficient of 0.901. Exploratory factor analysis revealed a cumulative variance explanation rate of 68.3%, confirming the scale’s good structural validity.

#### 2.3.3. Self-Control Questionnaire

The Chinese version of the Brief Self-Control Scale (*BSCS*), revised by [43] ([43]) and based on the work of [62] ([62]), was employed in this study. This 7-item scale assesses two dimensions: self-discipline and impulse control (e.g., *“I sometimes cannot resist doing things I know are wrong”*). Responses are rated on a 5-point Likert scale, ranging from 1 (*strongly disagree*) to 5 (*strongly agree*). The scale has demonstrated robust psychometric properties in Chinese samples (*Cronbach’s α* = 0.89–0.93) ([38]; [36]). In this investigation, internal consistency reliability reached 0.914. Confirmatory factor analysis confirmed strong alignment between the two-factor structure and theoretical model (*TLI* = 0.931, *SRMR* = 0.045), indicating notable cross-sample transportability of the instrument.

#### 2.3.4. Occupational Deviance Questionnaire

The University Counselors’ Occupational Deviance Questionnaire, developed by [68] ([68]), was employed in this study. This 11-item instrument assesses four dimensions: negligence of duties, demeaning or corporal punishment, abuse of power for personal gain, and unfair rewards or punishments. It utilizes a 5-point Likert scale, where a score of 1 indicates “absolutely would not” and a score of 5 indicates “absolutely would”, with higher scores reflecting an increased risk of occupational deviance. The questionnaire is well established in educational research and possesses validated psychometric properties (*Cronbach’s α* = 0.85–0.89) ([73]; [65]). In the current study, the instrument exhibited strong internal consistency reliability (*α* = 0.883). Confirmatory factor analysis indicated a strong model fit (*CFI* = 0.912, *RMSEA* = 0.068), confirming a robust theoretical alignment between the measurement structure and the proposed model.

To mitigate social desirability bias in occupational deviance items, the research team employed the measurement optimization strategy proposed by [64] ([64]). Utilizing a “distributed embedding” approach, the 11 occupational deviance items were divided into 4 subsets and integrated into the professional identity and self-control scales. This gradual exposure minimized psychological defensiveness and prevented the activation of concentrated defensive responses.

#### 2.3.5. Statistical Analyses

Data analysis was conducted using SPSS version 23.0, along with the AMOS plug-in. Statistical significance was set at α = 0.05 (*two-tailed*) to ensure analytical validity and reliability.

Data quality assessment: Before conducting formal analysis, the quality of the data was evaluated. The Kolmogorov–Smirnov (*K-S*) test was used to assess the assumptions of normal distribution ([18]). Concurrently, Harman’s single-factor exploratory factor analysis was performed to identify any common method bias ([56]). These procedures provided a solid foundation for the subsequent statistical analyses.

Descriptive statistics and correlation analysis: Descriptive statistics summarized key variables, with continuous data presented as M ± SD. To qualitatively describe variable levels within the sample, Z-tests against theoretical scale midpoints served as alternative benchmarks. This method facilitates the preliminary interpretation of questionnaire results based on theoretical frameworks in the absence of empirically calibrated sample data. Partial correlation analysis ([74]) was employed to examine relationships among core variables while controlling for demographic factors (*gender*, *age*, *academic rank*, *years of service*, *marital status*, *and student caseload*), thereby clarifying the intrinsic connections between professional identity, self-control, and occupational deviance.

Testing the moderating effect: To examine the moderating effect of counselors’ self-control on the relationship between professional identity and occupational deviance, hierarchical regression analysis was conducted using [26] ([26]) PROCESS macro, with occupational deviance as the dependent variable ([20]). The independent variable (*professional identity*) and the moderator (*self-control*) were mean-centered to create an interaction term “Professional Identity × Self-Control” in three steps: Step 1 included demographic control variables; Step 2 added the centered independent variable; and Step 3 introduced the interaction term. Bootstrap sampling with 5000 iterations was employed to calculate 95% confidence intervals, with Δ*R*^2^ recorded at each step to evaluate changes in the model’s explanatory power. For significant interactions, simple slope analysis categorized self-control at mean ±1 standard deviation. Visualization plots illustrated the varying predictive patterns of professional identity on occupational deviance across different levels of self-control, thereby clarifying the moderation mechanism.

## 3. Results

### 3.1. Normality Test and Common Method Bias Assessment

The Kolmogorov–Smirnov test was employed to assess the normality of the questionnaire data. The findings revealed absolute kurtosis values of 0.879, 0.507, and 1.059 for professional identity, self-control, and occupational deviance, respectively, as well as absolute skewness values of 0.152, 0.065, and 0.465. According to [18] ([18]), strict criteria for normality are seldom fulfilled in practical applications; however, data may be regarded as approximately normal if the absolute kurtosis is below 10 and the absolute skewness is below 3. Consequently, the data in this study met the criteria for approximate normality.

Given that all variables were self-reported, Harman’s single-factor test was employed to assess the presence of common method bias. The results demonstrated a chi-square-to-degrees-of-freedom ratio (*χ*^2^/*df*) of 3.586, which exceeds the acceptable threshold of less than 3. Additionally, the goodness-of-fit indices yielded the following results: Goodness-of-Fit Index (*GFI*) = 0.549, Comparative Fit Index (*CFI*) = 0.652, Normed Fit Index (*NFI*) = 0.614, and Non-Normed Fit Index (*NNFI*) = 0.683, all of which fall below the recommended standard of greater than 0.9. Furthermore, the Root Mean Square Error of Approximation (*RMSEA*) was calculated at 0.121, surpassing the threshold of less than 0.10, while the Root Mean Square Residual (*RMR*) was found to be 0.089, which significantly deviates from the acceptable standard of less than 0.05 ([29]). Factor analysis identified five factors with eigenvalues exceeding 1, which collectively accounted for 71.06% of the total variance. The largest common factor explained 31.89% of the variance, remaining below the critical threshold of 40% ([56]). These findings confirm that the data did not converge into a single factor, thereby indicating adequate discriminant validity and the absence of significant common method bias that could potentially influence the results.

### 3.2. Descriptive Analysis of Key Variables

The mean score for counselors’ professional identity was 3.447 ± 1.239, which is significantly greater than the theoretical median value of 3.00. The null hypothesis, which posited that the mean score is equal to the theoretical median, was rejected at the 95% confidence level (*z* = 6.874, *p* < 0.001). This finding suggests a relatively robust professional identity among the sample.

The mean score for self-control was 3.954 ± 0.971, which surpasses the theoretical median value of 3.00. The null hypothesis was also rejected (*z* = 18.719, *p* < 0.001), suggesting that counselors exhibit elevated levels of self-control.

Utilizing a cutoff of mean occupational deviance scores of ≥4, which corresponds to a response of “likely would” on the assessment scale, it was determined that 51 participants (14.05%) met the criteria for occupational deviance. The detection rate was computed as (*number of detected cases*/*valid sample size*) × 100%. The overall mean occupational deviance score was 2.553 ± 1.230, which is significantly lower than the theoretical median value of 3.00. The null hypothesis, which posited that “the mean score equals the theoretical median, was rejected at the 95% confidence level (*z* = −6.9149, *p* < 0.001). These findings suggest that, although instances of occupational deviance do occur among counselors, the overall prevalence remains relatively low.

### 3.3. Partial Correlation and Collinearity Analysis

After controlling for potential confounding variables—including gender, age, years of service, marital status, student load, and professional title—a covariate-adjusted partial correlation analysis was conducted to examine the intrinsic associations among key variables.

As illustrated in Table 2, the findings indicated a significant negative correlation between professional identity and occupational deviance (*r* = −0.599, *p* < 0.01), a significant positive correlation between professional identity and self-control (*r* = 0.391, *p* < 0.01), and a significant negative correlation between self-control and occupational deviance (*r* = −0.472, *p* < 0.01).

Research has demonstrated that a correlation coefficient with an absolute value exceeding 0.8 indicates a substantial likelihood of collinearity issues among variables ([71]). In the present study, while the correlation coefficients among the variables remained below this critical threshold, collinearity was further evaluated using the Variance Inflation Factor (*VIF*) criteria to uphold methodological rigor. In the context of collinearity analysis, VIF values exceeding 10 or tolerance levels falling below 0.1 are indicative of problematic collinearity ([9]). The results, as presented in Table 2, revealed the following VIF values: professional identity (4.954), self-control (2.826), and occupational deviance (4.171), all of which are below the threshold of 10. Correspondingly, the tolerance values were 0.194, 0.354, and 0.253, all exceeding 0.1. These findings substantiate the conclusion that significant collinearity issues are absent in the data.

### 3.4. Moderating Effect Analysis of Self-Control

Table 3 presents regression results for the effects of professional identity, self-control, and their interaction term on occupational deviance. The results demonstrate the following: Professional identity significantly negatively predicted occupational deviance (*Model 3: β* = −0.477, *p* < 0.001), indicating that a 1 SD increase in professional identity corresponded to a 0.477 SD decrease in occupational deviance. Self-control exerted a significant negative moderating effect on the professional identity-occupational deviance relationship (*β* = −0.171, *p* < 0.001). This signifies that each 1 SD increase in self-control enhances the negative predictive effect of professional identity on occupational deviance by 0.171 SD, confirming that self-control strengthens the negative predictive effect of professional identity on occupational deviance.

Further analysis revealed the moderating effect accounted for 2.8% of variance in occupational deviance (Δ*R*^2^ = 0.028). According to the criteria established by [70] ([70]) in the context of social science research, a Δ*R*^2^ value greater than 0.01 signifies that a moderating variable has a meaningful contribution to the model. When Δ*R*^2^ exceeds 2%, the moderation effect exhibits considerable explanatory power and possesses practical significance. Therefore, the moderation effect identified in this study is of actionable value for practical interventions.

Simple slope analysis results (Figure 1) revealed that at low levels of self-control, the negative predictive effect of professional identity on occupational deviance was weaker (*β* = −0.306, *p* < 0.001). Conversely, at high levels of self-control, this negative predictive effect significantly strengthened (*β* = −0.648, *p* < 0.001). This indicates that the negative predictive effect of professional identity on occupational deviance intensifies as the levels of self-control increase.

## 4. Discussion

### 4.1. Occupational Deviance Among Counselors Is Observed to Occur at Relatively Low Levels

This study finds a low prevalence of occupational deviance among instructors in Chinese higher education, aligning with previous research by [73] ([73]), [36] ([36]), and [27] ([27]). This positive ethical climate may stem from the synergistic interaction of multiple factors operating through distinct yet complementary mechanisms.

Robust external constraints are vital for maintaining high standards in educational institutions. The rigorous selection process, guided by regulations like the Construction of Counselor Teams in Regular Higher Education Institutions of the People’s Republic of China ([49]), serves as an initial screening mechanism. Universities prioritize academic qualifications and include background checks, ideological assessments, and ethical evaluations ([15]; [25]; [42]). This process ensures that new professionals have a strong ethical foundation and political awareness. Continuous professional development further reinforces these standards through regular training sessions and specialized workshops ([14]; [36]). A multi-dimensional oversight system, incorporating student feedback, peer review, and administrative assessments ([22]; [40]), helps identify potential ethical deviations early ([35]). Clearly defined penalties for violations ([57]) serve as a deterrent, ensuring accountability.

The internalization of competencies is crucial for personal autonomy and professional development, and the realization of its core role is inseparable from systematic capacity-building mechanisms in educational practice: Continuous participation in specialized training on educational psychology and student management methods helps educators deepen their understanding of students’ psychological needs and developmental stages ([40]). This in-depth grasp of students’ needs is not only an accumulation of professional skills but also gradually fosters a strong sense of professional identity, as evidenced by the high level of identity among teachers in this study. A strong professional identity further translates into internalized responsibility: it prompts educators to clearly recognize the profound impact of their behaviors on students’ development, thereby proactively adhering to ethical standards ([49]). Meanwhile, proficiency in student management strategies can simultaneously enhance self-regulatory abilities ([73]), which are particularly crucial when dealing with challenging situations and maintaining professional composure. Ultimately, professional identity and self-regulation form a dynamically complementary safeguard mechanism: The former provides intrinsic motivation for ethical behavior, while the latter offers practical tools for behavioral choices in complex situations. Together, they constitute a solid guarantee against deviant behaviors.

### 4.2. Professional Identity Negatively Predicts Occupational Deviance

This study indicates that counselors’ strong professional identity negatively predicts occupational deviance, with its intrinsic mechanisms systematically elucidated through the core framework of Social Cognitive Theory ([6]).

Counselors with a strong professional identity have clear and positive self-concepts ([77]). They recognize that their actions influence both their individual identities and the broader perception of the counseling profession’s role in student development. By seeing themselves as essential facilitators, they prioritize professional standards ([73]). Conversely, those with a weak professional identity lack self-awareness, leading to an unclear understanding of their role and a disregard for professional norms ([77]).

Social Learning Perspective indicates that counselors with a strong professional identity observe and emulate exemplary peers, internalizing effective practices ([6]). They are motivated by positive role models to improve their skills. In contrast, those with a weak professional identity show little interest in learning from others, making them more susceptible to negative influences and misconduct ([59]).

Self-efficacy theory, a key aspect of Social Cognitive Theory, highlights the connection between professional identity and behavioral compliance ([32]). Individuals with a strong professional identity develop high self-efficacy through mastery and vicarious experiences, enhancing their behavioral controllability in ethical dilemmas ([73]). Conversely, low self-efficacy can lead counselors to abandon professional standards under pressure, increasing the risk of violations ([30]).

According to Social Cognitive Theory’s triadic reciprocal determinism, counselors with a strong professional identity engage positively with their environment ([41]). They utilize resources like professional training and supervisory support for self-development and foster positive professional ecosystems. This mechanism helps them maintain consistent behavior despite negative stimuli ([58]). Conversely, those with low professional identity struggle with cognitive–environmental coordination, making them vulnerable to “deindividuated” behavior in adverse environments, weakening professional norms ([63]).

Social Cognitive Theory collectively elucidates the influence of professional identity on behavioral compliance ([6]). Individuals construct their professional identities cognitively, acquire behaviors through observational learning, sustain consistency via self-efficacy, and regulate their conduct within dynamic social interactions ([15]). This theoretical framework conceptualizes occupational deviance as emerging from the interaction between cognitive processes and social environments, thereby offering a foundation for comprehending counselors’ occupational behavior.

### 4.3. The Moderating Role of Self-Control in the Relationship Between Professional Identity and Occupational Deviance

This study demonstrates that self-control serves as a moderating variable in the relationship between counselors’ professional identity and instances of occupational deviance. Specifically, it enhances the inhibitory effect of professional identity on occupational deviance. This finding is consistent with the Conservation of Resources Theory ([28]), which posits that individuals are motivated to acquire, retain, and protect their valued resources. Within the context of the counseling profession, professional identity is identified as a crucial personal resource. Counselors who possess a strong professional identity invest cognitive, affective, and behavioral energy into their roles, thereby creating reserves of psychological energy ([35]). Self-control functions as a regulatory mechanism for resource management, allowing counselors to strategically allocate their professional identity resources when faced with stressors or temptations ([58]). For instance, when confronted with anger stemming from student misconduct, counselors with high levels of self-control can leverage their professional identity to manage negative emotions, reaffirm their roles as educators, and pursue constructive solutions, thus avoiding impulsive reactions ([52]). Consequently, self-control enhances the negative impact of professional identity on occupational deviance by mitigating the depletion of identity resources that may result from unethical behaviors ([73]).

Furthermore, self-control may act as a moderating factor in the relationship between professional identity and occupational deviance, particularly when combined with social support resources ([63]). In supportive educational environments, counselors benefit from the social backing of colleagues, supervisors, and institutional frameworks, which can enhance their self-control capabilities ([37]). For instance, when counselors perceive that they are understood and respected by their peers, they exhibit greater confidence and a stronger ability to exercise self-control when facing ethical challenges, thereby reinforcing the inhibitory effect of their professional identity on occupational deviance. At the same time, counselors with high levels of self-control are more inclined to actively seek out and utilize social support ([73]); they can articulate their professional dilemmas or stressors in an appropriate manner rather than resorting to occupational deviance as a form of emotional release, which, in turn, strengthens their professional identity and fosters a virtuous cycle.

In conclusion, this study employed empirical analysis to examine the prevalence of occupational deviance among university counselors, the negative predictive influence of professional identity on such deviance, and the moderating role of self-control within this relationship. The findings offer both a theoretical framework and empirical evidence for understanding the professional behaviors of counselors while also establishing a foundation for future research aimed at regulating professional conduct through the reinforcement of professional identity and the enhancement of self-control capacities.

## 5. Summary

### 5.1. Research Conclusions

This study empirically validates the relationship between university counselors’ professional identity and instances of occupational deviance, highlighting the moderating effect of self-control within this context. The key conclusions are as follows:(1)Occupational deviance among counselors is observed to occur at relatively low levels.(2)Counselors’ professional identity significantly and negatively predicts occupational deviance.(3)Self-control has a significant moderating effect on the negative relationship between professional identity and occupational deviance, enhancing the negative predictive relationship between these variables.

In summary, the findings clarify that professional identity plays a pivotal role in mitigating occupational deviance among university counselors, while self-control functions as a critical moderating variable that amplifies this inhibitory impact. These findings advance the theoretical comprehension of the mechanisms influencing counselors’ professional behaviors and offer practical implications for regulating such behaviors by promoting professional identity and cultivating self-control capacities. Furthermore, this study contributes to the literature on Social Cognitive Theory and Conservation of Resources Theory, providing both theoretical and empirical foundations for strengthening counselor teams and maintaining professional ethics.

### 5.2. Research Limitations

This study presents several methodological limitations and identifies potential avenues for enhancement.

(1) Sample structural limitations: The sample size of this study was relatively small, with participants primarily consisting of younger individuals and being concentrated within specific institutions and geographic regions. This sampling methodology may lead to an underestimation of population heterogeneity and restrict the generalizability of the findings to a wider array of counselor populations ([21]). Future research should implement stratified sampling techniques to develop participant distribution models that possess greater ecological validity, thereby ensuring improved representation of diverse demographic and institutional contexts ([60]).

(2) Validity constraints of research design: Although the cross-sectional design successfully captures static relationships among variables, it fails to clarify the dynamic evolutionary pathways that connect professional identity and occupational deviance ([48]). It is recommended that longitudinal tracking designs utilizing cross-lagged panel models be employed to uncover temporal associations between these variables. In particular, multi-year longitudinal studies (*with a tracking period of three years or more*) should be conducted to develop stage-specific models of professional identity development, with a focus on shifts in psychological mechanisms during critical career transitions, such as promotions ([17]).

(3) Measurement response bias: The sensitive nature of measures related to occupational deviance may elicit impression management tendencies among participants. Although the questionnaire employed indirect questioning techniques (e.g., *colleagues might demonstrate self-control when experiencing work-related stress*) and distributed 11 sensitive items across the professional identity and self-control modules, achieving complete mitigation of social desirability bias remains methodologically challenging ([56]). Future research should consider employing multi-source data triangulation, which includes informant reports (e.g., *evaluations by department leaders*) and Behavioral Event Interviews, to improve the accuracy of measurements ([34]).

(4) Theoretical model simplification: Although this study concentrated on the moderating role of self-control, the mechanisms by which professional identity affects occupational deviance likely involve multiple pathways. Future research should investigate additional mediating and moderating variables, such as occupational emotions ([16]), psychological resilience ([39]), and social support ([12]), to reveal more nuanced explanatory frameworks. This approach would establish a more comprehensive theoretical foundation for developing evidence-based strategies aimed at strengthening university counseling teams.

### 5.3. Research Implications

This study empirically establishes professional identity as a core agentic factor that significantly and negatively predicts occupational deviance among university counselors, addressing the traditional institutional oversight of over-relying on external constraints. Simultaneously, it reveals the enhancing effect of self-control on this predictive relationship, achieving an organic integration of Social Cognitive Theory ([6]) and Conservation of Resources Theory ([28]). These findings lay the theoretical foundation for a novel governance paradigm termed “institutional-psychological synergy” ([10]).

Based on the inhibitory effect of counselors’ professional identity on occupational deviance, both university administrators and individual counselors can implement strategies to strengthen professional identity and reduce deviant behaviors.

For administrators, in recruitment and selection, prioritize assessing candidates’ professional identity as a key hiring criterion to ensure that new counselors start with a strong foundation ([41]). When refining training systems, integrate content that strengthens identity, such as career development planning courses and structured experience-sharing sessions ([73]). Within incentive mechanisms, offer both material and symbolic rewards to counselors who demonstrate a high professional identity and ethical conduct, while incorporating these metrics into evaluation systems ([58]). Additionally, cultivate psychologically healthy work environments by reducing stressors (e.g., *excessive paperwork*) and fostering value-driven organizational cultures. For individual counselors, develop self-awareness to deepen professional understanding through pedagogical training, enhancing identity and resisting deviance ([15]). Recognize the critical role of self-regulation and emotion management in maintaining ethical standards, and acquire evidence-based techniques for emotion regulation. In career planning, align professional goals with identity commitments to facilitate growth. To maintain relationships, build constructive communication networks with stakeholders to reinforce identity and inhibit deviance ([16]).

Since self-control enhances the inhibitory effect of professional identity on occupational deviance, several additional implications arise. For administrators, it is essential to assess candidates’ self-control capacities during the selection and onboarding processes and to incorporate relevant training modules ([27]). In the context of professional development, self-control enhancement should be integrated into personalized career planning through targeted workshops ([73]). Furthermore, cultivating environments that promote self-regulation can be achieved by implementing value-aligned policies and practices. Establishing monitoring and feedback mechanisms will help identify issues promptly and provide customized guidance ([30]). For individual counselors, developing self-awareness is crucial in recognizing the critical role of self-control in sustaining professional identity and ethical conduct. Counselors should identify their personal vulnerabilities and incorporate techniques for fortifying their identity ([73]). Continuous learning and self-monitoring are vital; this can be accomplished by acquiring evidence-based self-control strategies, establishing behavioral tracking systems, and adjusting approaches when early indicators of deviance are detected ([15]). When faced with challenges, counselors should employ impulse-delay tactics to maintain their professional identity under pressure, thereby strengthening the alignment between their identity and compliance through enhanced self-control ([43]).

## Figures and Tables

**Figure 1 behavsci-15-01278-f001:**
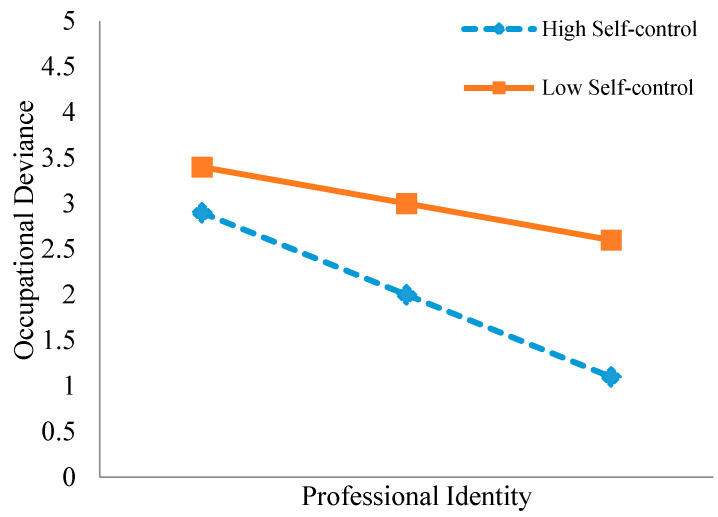
Slope graph of the moderating effect of self-control.

**Table 1 behavsci-15-01278-t001:** The demographic characteristics of the participants in each group (*n* = 363).

Demographic Variables	Form	Number (*n*)	Percentage (*%*)
Gender	Male	153	42.15
	Female	210	57.85
Age ^①^	<30 years old	169	46.56
	30–39 years old	152	41.87
	40–49	36	9.92
	≥50 years old	6	1.65
Marital status	Yes	237	65.29
	No	126	34.71
Years of service	<5 years	148	40.77
	5–10 years	113	31.13
	11–15 years	63	17.36
	>15 years	39	10.74
Professional title ^②^	Assistant	57	15.70
	Lecturer	230	63.36
	Associate Professor	70	19.28
	Professor	6	1.66
Student caseload ^③^	<200 students	57	15.70
	200–300 students	78	21.49
	>300 students	228	62.81
Add up the total		363	100

***Note:*** ① The high proportion of young faculty in China’s university counselor workforce stems from two main reasons: First, the job demands high intensity, requiring counselors to handle complex tasks, work night shifts, and respond to emergencies—pressures that young professionals are generally better equipped to manage. Second, universities position this role as a transitional stage for developing young faculty and administrative cadres; after gaining grassroots experience, these counselors often transition to other positions, resulting in higher mobility. ② Within the Chinese higher education system, university counselors are categorized as technical professionals within academic personnel structures. Their academic ranking system corresponds to that of full-time faculty members and consists of four levels: Assistant, Lecturer, Associate Professor, and Professor. Among these, the titles of “Associate Professor” and “Professor,” due to the long-term stability following appointment, can be regarded as having characteristics similar to “tenured positions.” However, within the counselor role, this group often possesses extensive experience in student affairs along with certain academic guidance capabilities, frequently undertaking responsibilities such as team management and the training of young counselors. They constitute a core force within the workforce that combines stability and leadership. ③ The student caseload of counselors refers to the number of students for whom they are responsible for comprehensive administrative management, excluding those they teach.

**Table 2 behavsci-15-01278-t002:** Partial correlation and collinearity analysis of key variables (*n* = 363).

	M ± SD	1	2	3	Collinearity Analysis	
					VIF Value	Tolerance
1. Professional identity	3.447 ± 1.239	1			4.954	0.194
2. Self-control	3.954 ± 0.971	0.391 **	1		2.826	0.354
3. Occupational deviance	2.553 ± 1.230	−0.509 **	−0.412 **	1	4.171	0.253

***Note:*** ** *p* < 0.01.

**Table 3 behavsci-15-01278-t003:** Analysis of the moderating effect of self-control (n = 363).

	Model 1					Model 2					Model 3				
	*B*	SE	*t*	*p*	*β*	*B*	SE	*t*	*p*	*β*	*B*	SE	*t*	*p*	*β*
Constant	1.870	0.224	8.3420 ***	0.000	-	1.787	0.224	7.982 **	0.000	-	1.458	0.210	6.942 ***	0.000	-
Gender	0.043	0.051	0.844	0.399	0.021	0.059	0.051	1.165	0.245	0.029	0.092	0.047	1.953	0.052	0.045
Age	0.022	0.044	0.500	0.618	0.016	0.019	0.043	0.437	0.662	0.014	0.043	0.040	1.081	0.280	0.031
Years of service	−0.017	0.029	−0.565	0.572	−0.017	−0.009	0.029	−0.295	0.768	−0.009	0.016	0.027	0.597	0.551	0.016
Marital status	0.006	0.065	0.092	0.926	0.003	0.010	0.064	0.148	0.883	0.005	0.061	0.060	1.020	0.308	0.029
Professional title	0.023	0.021	1.097	0.273	0.026	0.023	0.020	1.122	0.263	0.026	0.019	0.019	1.025	0.306	0.022
Student caseload	0.006	0.030	0.210	0.834	0.005	0.022	0.030	0.740	0.460	0.019	0.025	0.028	0.896	0.371	0.021
Professional identity	−0.407	0.023	−18.626 **	0.000	−0.404	−0.491	0.038	−6.059 **	0.000	−0.488	−0.480	0.035	−7.980 **	0.000	−0.477
Self-control						−0.109	0.039	−2.810 **	0.005	−0.109	−0.111	0.036	−3.125 **	0.002	−0.111
Professional identity × Self-control											−0.154	0.019	−8.062 **	0.000	−0.171
*R* ^2^	0.815					0.819					0.847				
Adjusted *R*^2^	0.811					0.815					0.843				
*F*	*F* (7,355) = 223.075, *p* = 0.000					*F* (8,354) = 199.968, *p* = 0.000					*F* (9,353) = 217.105, *p* = 0.000				
Δ*R*^2^	0.815					0.004					0.028				
Δ*F*	*F* (7,355) = 223.075, *p* = 0.000					*F* (1,354) = 7.895, *p* = 0.005					*F* (1,353) = 64.995, *p* = 0.000				

***Note:*** The dependent variable is occupational deviance. ** *p* < 0.01, *** *p* < 0.001.

## Data Availability

The data supporting the findings of this study are available from the corresponding author upon reasonable request: https://orcid.org/0000-0002-9183-411X.

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
