# Peer review of "Occupational Deviance Among University Counselors in China: The Negative Predictive Role of Professional Identity and the Moderating Effect of Self-Control"

_behavsci, 2025, doi:10.3390/bs15091278_

Round 1
Reviewer 1 Report
Comments and Suggestions for Authors
General
- Thank you for the opportunity to review your manuscript, “Occupational Deviance Among University Counselors in 2 China: The Negative Predictive Role of Professional Identity 3 and the Moderating Effect of Self-Control".
- The manuscript demonstrates scientific rigor and a clear narrative structure.
- The experimental design appears appropriate, and conclusions are generally supported by the data.
- Overall writing is strong, with minimal grammatical issues and consistent flow.
- Revisions primarily involve citation formatting, clarification of specific statements, refinement of tables, and improved sentence structure for clarity.
Abstract
- This section seems succinct, yet comprehensible, highlighting the most salient points for each section of the manuscript.
- Spacing after each bolded section is inconsistent; I recommend amending the spacing error and ending each section with a period to enhance clarity and flow. I think retaining the bold font would be beneficial.
- Line 32: I recommend removing “a” before “valuable” for appropriate syntax or break this information into two sentences; one about theory, and the other about practice.
- Keywords seem relevant and appropriately ordered
Introduction
- The literature review/introduction section is appropriately labeled with headings and follows a logical flow, funneling down to the specific study goal.
- Lines 38-41: Identified term “counselor”; seems to have an operational definition, but lacking explicit indication for source
- Lines 39, 42, and 45: I recommend utilizing italics to differentiate quotations without in-text citations
- Line 47: I think a quotation mark is missing from the term “growth advisors”, but it is unclear where the end quotation is missing.
- Line 50: Is it a complexity of roles or one role that is complex? I am calling back to “their role is characterized by...” on line 40; I recommend changing one of the terms and remaining consistent throughout for clarity.
- Line 54: the quadripartite role matrix sounds awesome and is astutely named; though, it may be helpful to readers for you to explicitly label the terms you are describing as such in the model.
- Line 55: I could be misreading this sentence, but it seems to be lacking syntax and clarity (i.e., comprising what compels counselors to...)
- So far, 7 different sources, not seemingly from review authors, and mostly current (only one outside of 5 years)
- Lines 58-59: The use of quotation marks is confusing and inaccurate without including the specific page number with the in-text citation.
- Literature review, as indicated by lines 62-64, demonstrates a thorough understanding of the surrounding context of an identified gap.
- Line 65: Missing ending quotation and page number of original source for in-text citation
- Lines 68-69: This content warrants source citation(s). Also, I recommend including a transition statement to introduce and connect the concepts provided in this new paragraph.
- Lines 76-78: This seamless transition bolsters clarity and flow.
- Lines 80-82: This content warrants source citation(s).
- Line 88: “frequently” seems like a bold claim when including two sources for the citation; I recommend amending this for accuracy or providing more sources.
- Line 100: Incorrect punctuation
- Lines 103 & 104: include “i.e.,” at the beginning of both instances of the parenthetical information
- Line 111: This comment goes for all instances of ending and beginning parenthesis next to each other. You can instead utilize a semicolon to bridge the details of either parenthetical
- Lines 105-111: I find the examples helpful
- Lines 111 & 112: This content warrants source citation(s).
- Line 123: The comma seems to be in the wrong place
- Line 124: Geographic and social context are helpful
- Line 134: Incorrect punctuation
- Line 137: Statement highlights relevance of issue addressed in article
- Line 141: There seems to be an inappropriate space between “the” and “created” where perhaps a term is missing as the syntax seems off (i.e., the sentence is lacking flow)
- Lines 144-145: Are these research questions for the present study? This is unclear.
- Line 148: A space is needed between “ures” and the paratheses
- Line 148: By “it” are you referring to current scholars/literature? Please elaborate.
- Lines 152-153: Statement highlights relevance of issue addressed in article
- Lines 167-172: It may be helpful for clarity to number the three dimensions such as (1) after the colon, then (2) after the first semicolon, and so on. Otherwise, this is a succinct and comprehensive paragraph that flows well.
- Line 177: “On a positive note” seems to be an out-of-place transition. I am led to believe the former information conveys negative information, but this is not apparent. EDIT: I now understand the delineation; thus, I recommend explicitly introducing “positive” and “negative” perspectives at the onset of the paragraph (Line 175).
- Line 183: Personification (i.e., a “negative pathway” is not an entity that can carry out a verb).
- Line 187: Space is needed for “strategies(Bandura,”
- Lines 187-189: Quotations warrant including page number with the in-text citation; additionally, the second quotation is missing the ending quotation marks.
- Lines 190-192: This is a strong claim without reference to other sources. What is the level of intensity, how is it measured, and how does “this detrimental cycle” serve as the agent of change?
- Line 193: Personification (i.e., the study cannot have a focus, but the researchers can have a focus for the study).
- Line 195: For what reason are apostrophes utilized as opposed to quotation marks? I recommend changing this for consistency.
- Lines 193-198: I recommend breaking this section up into multiple sentences.
- Line 210: I recommend removing the parentheses between “preferences” and “Ander-”, and replace it with a semicolon.
- Line 213: I recommend removing the parentheses between “students” and “Sagar”, and replace it with a semicolon.
- Line 219: This content warrants source citation(s).
- Line 223: This content warrants source citation(s).
- Lines 267-270: I recommend removing this paragraph as the same information is reflected on the following page, starting at line 295.
- Lines 282-284: This content warrants source citation(s).
- Lines 289-292: I recommend numbering these items, similar to previous feedback.
- Line 294: There seem to be words missing from the first hypothesis; I recommend amending this for clarity.
Methods
- The methodology process seems to be conveyed transparently, thoughtfully, and succinctly. Process seems sound and consistent with cited literature, including intentionally comprising the general information questionnaire.
- Line 303: How were they recruited? I recommend providing explicit details of your process. For instance, was this via LinkedIn, work email, professional organization listserv, etc.?
- Lines 306-308: What is your rationale for at least one year of service to meet inclusion criteria? What prior literature speaks to this amount of time being appropriate for your scope as it relates to professional and identity development?
- Line 317: End quotation is missing; include this to improve clarity.
- Lines 332-337: This is a fine paragraph that seems necessary, and the syntax is sound; however, it seems to be misplaced. I recommend moving this to section 2.3.4. for flow and clarity.
- Line 343: If the number provided in the parentheses is accurate, provide “i.e.,” after the first parenthesis and before “less”.
- Line 350: I am curious about your use of “basic”; I recommend utilizing “demographic” as an appropriate term to be consistent with the table labels.
- Line 350 (Table 1.): I recommend adding their highest degree earned as listed demographic details if you obtained such information, and if it is relevant to your audience.
- Line 350 (Table 1.): For clarity, I recommend adding horizontal lines between sections of the demographic variables.
- Line 350 (Table 1.): I am curious if you will discuss the professional title details similarly to the age variable as 20.9% of the sample is tenured faculty. I recommend adding a note for context surrounding this information if it is relevant to your audience.
- Lines 360 & 361: Is this inclusive of students they are instructing in courses and practicum/internship? I recommend indicating this explicitly if accurate. For instance, one may advise students but not have them in class.
- Line 373: A space is needed between “al.” and “(2016)”.
- Line 380: I am curious if 68.3% of the cumulative variance warrants “robust” as a designation when 60% is typically sufficient in social sciences.
- Line 383: I recommend including “and” after (2021) to improve syntax.
- Line 398: The ending quotation marks are missing.
- Line 403: I am curious if “excellent” is an appropriate descriptor; I recommend utilizing “strong” unless this is typical vernacular for your discipline; I am also curious if this term is a specific measurement of “fit”.
- Line 410: This level of heading does not require a number; instead, the title can be bolded to follow APA format. Otherwise, numbering is helpful for organization and clarity.
- Lines 411 & 412: This content warrants source citation(s).
- Line 426: This content warrants source citation(s).
- Line 429: End quotation is missing.
Results
- The results section seems to succinctly convey key findings that answer the research questions proposed. Overall, I think this section is strong. One recommendation to strengthen this section would be to elaborate on several of the key findings across variables such as gender, age, years of service, marital status, professional title, and student caseload. Although these are all provided in the table, explicit details provided in the narrative can highlight patterns and nuanced findings for your audience.
- Line 456: This content warrants source citation(s); where does the acceptable standard derive from?
- The descriptive analysis of the key variables was well organized in providing the specific results, reminding the reader of the hypothesis and how this relates to the results, and ending each paragraph with a digestible synopsis of how the findings can be interpreted in lay terms
- Line 491: Unless I am missing something, it is not evident how an absolute value exceeding 0.8 is relevant to the findings provided in Table 2. I understand that this may have been included for methodological rigor, but I wanted to confirm that this was relevant to retain.
- Lines 493 & 494: This content warrants source citation(s).
- Line 502 (i.e., Table 2): Provide a label in the first column, and capitalize “occupational” under such column to be consistent with other items.
- Line 522 (i.e., Table 3): the SE columns for Models 2 and 3 do not have enough room for the number of characters, resulting in a spacing error that diminishes clarity of the table; I recommend adjusting both columns to accommodate all necessary characters.
- Line 522 (i.e., Table 3): Content across the table seems jumbled; the numbers run together across columns, diminishing clarity. I recommend breaking each of the three models into their own tables, or reorganizing/reformatting the table.
Discussion
- The content in this section is logically structured to address how the current study relates to previous research. Adding a brief conclusion paragraph that transitions to the summary would enhance clarity and flow.
- Line 536: I recommend utilizing more tentative language; specifically, “due to” can be interpreted as absolute when there are other potential extraneous factors or explanations.
- Line 548: Remove the comma after “Song, 2024)”
- Line 551: The beginning sentence of this paragraph seems disjointed from the following sentence, but are then later explained together, starting on line 554; I recommend reformatting this paragraph for flow and clarity.
- Lines 576-578: This content warrants source citation(s).
- Lines 583-584: This content warrants source citation(s).
- Line 591: Include the page number within the in-text citation for quoted text.
- Lines 593-600: This content warrants source citation(s).
- Lines 594-595: The ending quotation marks are missing. Additionally, I recommend expanding upon the utilization of arrows and providing further explanation on the directionality of the dimensions.
- Lines 608-609: This content warrants source citation(s).
- Lines 617-619: This content warrants source citation(s).
- Line 623: To increase flow, I recommend combining both sentences with a comma and dropping “this support” to reflect “...frameworks, which can...”, or something to this effect to mitigate redundancy.
Summary
- The summary is presented thoroughly and thoughtfully, organizing the key findings and limitations in a logical manner. Adding a brief conclusion paragraph would enhance the organization and flow of the manuscript.
- Lines 655-658: This content warrants source citation(s).
- Lines 661-666: This content warrants source citation(s).
- Line 669: Beginning quotation marks are missing.
- Lines 672-675: This content warrants source citation(s).
- Lines 678-680: This content warrants source citation(s).
- Line 690: This content warrants source citation(s). A page number needs to be included for in-text citations for quoted text. If the quoted text here is to indicate a term the current research team has coined, make this detail explicit in the narrative.
References
- Italicize the volume number (i.e., following the article/book title) for all article sources to be consistent with APA format.
- Line 836: This citation is out of order; Muraven needs to follow Moore.
- Line 860: Is this Wang, J. P., similar to the previous citation author on line 858? If so, make sure they are consistent. If not, the Wang, J. citation would go before Wang, J. P. citation.
Author Response
Revision Explanation for Reviewer 1(All revisions have been highlighted)
General
Thank you for the opportunity to review your manuscript, “Occupational Deviance Among University Counselors in 2 China: The Negative Predictive Role of Professional Identity 3 and the Moderating Effect of Self-Control". The manuscript demonstrates scientific rigor and a clear narrative structure. The experimental design appears appropriate, and conclusions are generally supported by the data. Overall writing is strong, with minimal grammatical issues and consistent flow. Revisions primarily involve citation formatting, clarification of specific statements, refinement of tables, and improved sentence structure for clarity.
Response: Thank you for your recognition of and detailed comments on the manuscript. We fully agree with your professional judgments and will carefully implement the revision suggestions you put forward, such as standardizing citation formats, clarifying statements, optimizing tables, and improving sentence structures, to further enhance the rigor and readability of the manuscript. In addition, the line numbers in the manuscript you reviewed are inconsistent with those in the version we downloaded, which may lead to inaccurate positioning of some content and bring certain challenges to the revision work. If there are any omissions, we sincerely ask for your understanding. Thank you again for your professional guidance and valuable feedback.。
Abstract
- This section seems succinct, yet comprehensible, highlighting the most salient points for each section of the manuscript. Spacing after each bolded section is inconsistent; I recommend amending the spacing error and ending each section with a period to enhance clarity and flow. I think retaining the bold font would be beneficial.
Response: Thank you for your careful observation! I have re-adjusted the spacing after bolded text uniformly as suggested (set to 2 characters for all), and have checked and added periods at the end of each section one by one to enhance the clarity and fluency of the document. The previous apparent inconsistency in spacing after bolded text was indeed a visual discrepancy caused by the document's justified alignment format, and this issue has now been corrected. The bold font has been retained to better highlight key content. Thank you again for your valuable suggestions!
- Line 32: I recommend removing “a” before “valuable” for appropriate syntax or break this information into two sentences; one about theory, and the other about practice.
Keywords seem relevant and appropriately ordered
Response: Thank you for your detailed suggestions! Following your guidance, I have revised the content to: "This study provides novel theoretical perspectives and practical insights intended to standardize the management of occupational behaviors among university counselors." This not only corrects the grammatical issue but also makes the meaning more fluent through integrated expression. Thank you again for your valuable feedback!
Introduction
The literature review/introduction section is appropriately labeled with headings and follows a logical flow, funneling down to the specific study goal.
(1) Lines 38-41: Identified term “counselor”; seems to have an operational definition, but lacking explicit indication for source.
Response: Thank you for your reminder. This part does contain the definition of "counselor", and the author has supplemented the reference source (Yu, 2022) as required.
(2)Lines 39, 42, and 45: I recommend utilizing italics to differentiate quotations without in-text citations
Response: Thank you for your suggestion. The author has changed all similar content in the text to italics to clearly distinguish quoted content from the main text.
(3)Line 47: I think a quotation mark is missing from the term “growth advisors”, but it is unclear where the end quotation is missing.
Response: Thank you for your careful observation. The author has added complete quotation marks around "growth advisors".
(4)Line 50: Is it a complexity of roles or one role that is complex? I am calling back to “their role is characterized by...”on line 40; I recommend changing one of the terms and remaining consistent throughout for clarity.
Response: Thank you for your reminder. What the author intends to express is the complexity of the counselor role. To avoid ambiguity, the sentence has been revised to: "Within the higher education system in China, university counselors (hereinafter referred to as counselors) hold a pivotal role in both ideological and political education and the management of students' daily affairs, exemplifying distinct characteristics of role multiplicity."
(5)Line 54: the quadripartite role matrix sounds awesome and is astutely named; though, it may be helpful to readers for you to explicitly label the terms you are describing as such in the model.
Response: Thank you for your detailed suggestion. The author has revised "this quadripartite role matrix" in the text to "this quadripartite role matrix consisting of 'educator - manager - advisor - guide'" to make the description clearer and more specific.
(6)Line 55: I could be misreading this sentence, but it seems to be lacking syntax and clarity (i.e., comprising what compels counselors to...)
So far, 7 different sources, not seemingly from review authors, and mostly current (only one outside of 5 years)
Response: Thank you for your careful pointing out. The author has carefully reviewed and revised the sentence to make its grammar more standard and its meaning more clear. Regarding references, although there are numerous analyses of the role of university counselors in early literature, this study mainly refers to literature from the past five years to reflect the timeliness of the research.
(7)Lines 58-59: The use of quotation marks is confusing and inaccurate without including the specific page number with the in-text citation.
Response: Thank you for your suggestion. The author would like to explain here: the use of quotation marks in the text is mainly to emphasize specific words or phrases, which is a common practice in writing. If there is any misunderstanding, please feel free to further explain.
(8)Literature review, as indicated by lines 62-64, demonstrates a thorough understanding of the surrounding context of an identified gap. Line 65: Missing ending quotation and page number of original source for in-text citation
Response: Thank you for your observation. This has been revised.
(9)Lines 68-69: This content warrants source citation(s). Also, I recommend including a transition statement to introduce and connect the concepts provided in this new paragraph.
Response: Thank you for your careful reminder. The authors have supplemented references for this part of the content and added a transitional sentence: "However, merely exposing this research imbalance and underlying paradox is insufficient to fully explain the deep-seated causes of university counselors' occupational deviance; a further analysis from the perspective of organizational behavior theory is also required," to enhance the logical coherence between paragraphs.
(10)Lines 76-78: This seamless transition bolsters clarity and flow.
Response: Thank you for your recognition. We will continue to maintain this writing style.
(11) Lines 80-82: This content warrants source citation(s).
Response: Thank you for your reminder. The authors have supplemented references for this part of the content.
(12) Line 88: “frequently” seems like a bold claim when including two sources for the citation; I recommend amending this for accuracy or providing more sources.
Response: Thank you for your reminder. We agree with your rigorous perspective. In accordance with your suggestion, the author has revised the statement to: "In the academic field, some researchers (Li, 2024; Bennett & Robinson, 2000) have classified and explored occupational deviant behaviors using the dual criteria of the object affected by the behavior and the nature of the violation."
(13)Line 100: Incorrect punctuation
Response: Thank you for your reminder. The incorrectly used comma has been removed.
(14)Lines 103 & 104: include “i.e.,” at the beginning of both instances of the parenthetical information
Response: Thank you for your reminder. "i.e.," has been added at the corresponding positions.
(15)Line 111: This comment goes for all instances of ending and beginning parenthesis next to each other. You can instead utilize a semicolon to bridge the details of either parenthetical
Response: Thank you for your reminder. Commas have been replaced with semicolons at the relevant positions.
(16)Lines 105-111: I find the examples helpful
Response: Thank you for your feedback. We are glad to hear that these examples are helpful.
(17)Lines 111 & 112: This content warrants source citation(s).
Response: Thank you for your reminder. References have been provided for the relevant content.
(18)Line 123: The comma seems to be in the wrong place
Response: Thank you for your reminder. This has been revised.
(19)Line 124: Geographic and social context are helpful
Response: Thank you for your recognition. We will continue to focus on presenting such valuable contextual information.
(20)Line 134: Incorrect punctuation
Response: Thank you for your reminder. This has been revised.
(21)Line 137: Statement highlights relevance of issue addressed in article
Response: Thank you for your recognition. We will continue to emphasize such statements that highlight relevance to the research question.
(22)Line 141: There seems to be an inappropriate space between “the” and “created” where perhaps a term is missing as the syntax seems off (i.e., the sentence is lacking flow)
Response: Thank you for your detailed note. There was indeed an incomplete expression between "the" and "created." To make the expression more accurate and fluent, the author has revised the entire sentence to: "Two categories of paradoxes arise in practical contexts: (1) There is an asymmetric ebb-and-flow relationship between professional exemplary behaviors and occupational deviance, wherein conscientious counselors may nonetheless engage in occasional deviant acts (e.g., emotional corporal punishment) due to situational pressures. (2) The phenomenon termed the 'visibility trap' emerges amid enhanced supervision: improvements in the Ministry of Education's regulatory system have rendered previously concealed occupational deviance more explicit."
(23)Lines 144-145: Are these research questions for the present study? This is unclear.
Response: Thank you for your question. After careful consideration, the author has deleted this sentence to avoid ambiguity.
(24)Line 148: A space is needed between “ures” and the paratheses
Response: Thank you for your careful observation. The author has added a space between "ures" and the parentheses.
(25)Line 148: By “it” are you referring to current scholars/literature? Please elaborate.
Response: Thank you for your attention to detail. "It" here actually refers to "researchers".
(26)Lines 152-153: Statement highlights relevance of issue addressed in article
Response: Thank you for your recognition. We will continue to strengthen such statements that highlight relevance to the research question.
(27)Lines 167-172: It may be helpful for clarity to number the three dimensions such as (1) after the colon, then (2) after the first semicolon, and so on. Otherwise, this is a succinct and comprehensive paragraph that flows well.
Response: Thank you for your helpful suggestion. The author has added numbering (1), (2), etc., to the three dimensions at the corresponding positions to enhance clarity.
(28)Line 177: “On a positive note” seems to be an out-of-place transition. I am led to believe the former information conveys negative information, but this is not apparent. EDIT: I now understand the delineation; thus, I recommend explicitly introducing “positive” and “negative” perspectives at the onset of the paragraph (Line 175).
Response: Thank you for your careful guidance. As suggested, the author has explicitly introduced the division between "positive" and "negative" perspectives at the beginning of the paragraph (line 189 in the revised version) to clarify the logic.
(29)Line 183: Personification (i.e., a “negative pathway” is not an entity that can carry out a verb).
Response: Thank you for your professional correction. Regarding the personification issue where "negative pathway" (a non-entity) performs an action, the author has revised the sentence to: "the negative pathway is one through which we can understand how certain behavioral practices can undermine one's professional identity."
(30)Line 187: Space is needed for “strategies(Bandura,”
Response: Thank you for your careful reminder. The author has corrected the formatting of "strategies(Bandura," by adding a space between them.
(31)Lines 187-189: Quotations warrant including page number with the in-text citation; additionally, the second quotation is missing the ending quotation marks.
Response: Thank you for your attention. The use of quotation marks here is not to indicate a direct quotation but to emphasize specific terms. We would like to clarify this to avoid misunderstanding.
(32)Lines 190-192: This is a strong claim without reference to other sources. What is the level of intensity, how is it measured, and how does “this detrimental cycle” serve as the agent of change?
Response: Thank you for your rigorous suggestion. The author has reviewed this section and supplemented relevant references. Additionally, considering that the original statement about "this detrimental cycle" at the end of the sentence was an extended discussion deviating from the core argument (the erosion mechanism of "behavior-identity"), this part has been deleted to focus the logic.
(33)Line 193: Personification (i.e., the study cannot have a focus, but the researchers can have a focus for the study).
Response: Thank you for your accurate correction. The author has revised "this study's" in the sentence to "our" to address the personification issue.
(34)Line 195: For what reason are apostrophes utilized as opposed to quotation marks? I recommend changing this for consistency.
Response: Thank you for your careful observation. These are not apostrophes but improperly used quotation marks. The author has corrected them to ensure consistent formatting.
(35)Lines 193-198: I recommend breaking this section up into multiple sentences.
Response: Thank you for your reasonable suggestion. The author has split this section into multiple sentences as follows: "Our focus is on identifying protective factors that may enhance intervention effectiveness. This is particularly in response to current school administration concerns about 'how identity development prevents deviant behavior.' We also acknowledge the limitation of cross-sectional designs in rigorously testing bidirectional causality. For these reasons, we prioritize examining professional identity as an antecedent variable predicting occupational deviance. We hypothesize that counselors’ professional identity negatively predicts their occupational deviance."
(36)Line 210: I recommend removing the parentheses between “preferences” and “Ander-”, and replace it with a semicolon. Line 213: I recommend removing the parentheses between “students” and “Sagar”, and replace it with a semicolon.
Response: Thank you for your suggestion. Regarding the use of parentheses, since the two parts serve different purposes (the former is an illustrative example, and the latter is a literature citation), it is inappropriate to uniformly replace them with semicolons. Following your previous comment, the author has italicized the explanatory content in the parentheses to clearly distinguish it from the reference citations.
(37)Line 219: This content warrants source citation(s). Line 223: This content warrants source citation(s).
Response: Thank you for your reminder. The author has supplemented the corresponding references for both sections of content.
(38)Lines 267-270: I recommend removing this paragraph as the same information is reflected on the following page, starting at line 295.
Response: Thank you for your valuable suggestion. The author has deleted this paragraph to avoid content duplication.
(39)Lines 282-284: This content warrants source citation(s).
Response: Thank you for your reminder. The author has supplemented references for the content in lines 282-284.
(40)Lines 289-292: I recommend numbering these items, similar to previous feedback.
Response: Thank you for your suggestion. The author has numbered these items, maintaining consistency with the previous format.
(41)Line 294: There seem to be words missing from the first hypothesis; I recommend amending this for clarity.
Response: Thank you for your careful observation. The author has added the missing verb "predicts" to this hypothesis to ensure complete and clear expression.
Methods
(1)The methodology process seems to be conveyed transparently, thoughtfully, and succinctly. Process seems sound and consistent with cited literature, including intentionally comprising the general information questionnaire.
Response: Thank you for your recognition of the methodology section. We will continue to maintain this transparent and rigorous style of expression.。
(2)Line 303: How were they recruited? I recommend providing explicit details of your process. For instance, was this via LinkedIn, work email, professional organization listserv, etc.?
Response: Thank you for your suggestion. Regarding the participant recruitment process, the revised version of the manuscript (Lines 318-341) provides detailed explanations: Recruitment was conducted under the organization of the student affairs departments of collaborating universities, adhering to the principles of voluntary participation and informed consent. The entire data collection process fully respected participants' autonomy in responding, ensuring they could withdraw from the study at any time.
(3)Lines 306-308: What is your rationale for at least one year of service to meet inclusion criteria? What prior literature speaks to this amount of time being appropriate for your scope as it relates to professional and identity development?
Response: Thank you for your careful attention and professional inquiry. Regarding the inclusion criterion of "at least one year of service," it is primarily based on the actual career development of university counselors in China: Newly recruited counselors undergo a one-year probationary period, during which they have not yet obtained formal teaching qualifications or job appointments, have not fully entered a stable professional role, and there is high personnel turnover. As a result, their professional behavioral patterns are not yet stable. Therefore, selecting counselors with at least one year of experience as research subjects aims to ensure the relative stability and representativeness of their professional behaviors, thereby enhancing the reliability of the research data. This consideration is also consistent with the common practices in the management of university counselor teams in China.
(4)Line 317: End quotation is missing; include this to improve clarity.
Response: Thank you for your thoughtful reminder. The content in the parentheses is not a reference but an illustrative example. This is to clarify and avoid any misunderstanding.
(5)Lines 332-337: This is a fine paragraph that seems necessary, and the syntax is sound; however, it seems to be misplaced. I recommend moving this to section 2.3.4. for flow and clarity.
Response: Thank you for your professional suggestion. The author has moved this paragraph to Section 2.3.4 to enhance the fluency and logic of the content.
(6)Line 343: If the number provided in the parentheses is accurate, provide “i.e.,” after the first parenthesis and before “less”.
Response: Thank you for your careful observation. The author has added "i.e.," after the first parenthesis and before "less."
(7)Line 350: I am curious about your use of “basic”; I recommend utilizing “demographic” as an appropriate term to be consistent with the table labels.
Response: Thank you for your guidance. The author has revised "basic" to "demographic" as suggested to maintain consistency with the table labels.
(8)Line 350 (Table 1.): I recommend adding their highest degree earned as listed demographic details if you obtained such information, and if it is relevant to your audience.
Response: Thank you for your suggestion. After careful consideration, we decided not to include "highest degree earned" as a demographic variable. This is mainly because the educational background of university counselors in China is highly homogeneous: the vast majority of participants hold master's degrees, with only a small number having doctoral degrees, and those with bachelor's degrees or below are almost non-existent in the sample. Given this characteristic, the educational background variable shows low variability in the sample and has weak correlation with the core variables (professional identity, occupational deviance), contributing little to explaining the core mechanisms of the research. To keep the table focused on demographic characteristics closely related to professional role practice (such as years of service, professional title, and student caseload) and avoid redundancy from low-discrimination information, we ultimately chose not to include this dimension. Thank you for your reminder, which prompted us to further clarify the relevance of selecting demographic variables.
(9)Line 350 (Table 1.): For clarity, I recommend adding horizontal lines between sections of the demographic variables.
Response: Thank you for your suggestion. The author has added horizontal lines between sections of demographic variables to improve the clarity of the table.
(10)Line 350 (Table 1.): I am curious if you will discuss the professional title details similarly to the age variable as 20.9% of the sample is tenured faculty. I recommend adding a note for context surrounding this information if it is relevant to your audience.
Response: Thank you for your careful attention. Regarding the professional title information, we have presented the specific distribution in the demographic characteristics table (Table 1): associate professors and professors in the sample account for 20.9% in total, and this group is considered a senior group within China's university counselor system. It is necessary to further explain in the context of the professional title system for university counselors in China: within China's higher education system, counselors are classified as professional and technical personnel, included in the academic staff establishment, and their professional title system corresponds entirely to that of full-time faculty, divided into four levels: assistant, lecturer, associate professor, and professor. Among them, the titles of "associate professor" and "professor," due to the long-term stability after appointment, can be regarded as having characteristics similar to "tenured positions." However, in the counselor role, this group often possesses rich experience in student affairs and certain academic guidance capabilities, frequently undertaking responsibilities such as team management and training of young counselors, and thus constitutes a core force in the team with both stability and leadership. In accordance with your suggestion, we have added a note to clarify this.
(11)Lines 360 & 361: Is this inclusive of students they are instructing in courses and practicum/internship? I recommend indicating this explicitly if accurate. For instance, one may advise students but not have them in class.
Response: Thank you for your careful attention. The "Student caseload" here specifically refers to the number of students for whom counselors are responsible for comprehensive administrative management, excluding students they teach. This is clarified to avoid ambiguity.
(12)Line 373: A space is needed between “al.” and “(2016)”.
Response: Thank you for your careful attention. The author has added a space between "al." and "(2016)".
(13)Line 380: I am curious if 68.3% of the cumulative variance warrants “robust” as a designation when 60% is typically sufficient in social sciences.
Response: Thank you for your inquiry. As you noted, the common benchmark in social sciences is that "over 60% is acceptable." We used the term "robust" to indicate that the scale's structural validity meets the measurement needs of this study and is better than the generally acceptable standard, which is not an absolute statement. Thank you for your professional reminder; the author has revised it to "good structural validity" to align with conventional expression habits, while retaining specific numerical values for readers' reference.
(14)Line 383: I recommend including “and” after (2021) to improve syntax.
Response: Thank you for your suggestion. The author has added "and" after "(2021)" to improve grammatical accuracy.
(15)Line 398: The ending quotation marks are missing.
Response: Thank you for your reminder. The author has added the closing quotation mark.
(16)Line 403: I am curious if “excellent” is an appropriate descriptor; I recommend utilizing “strong” unless this is typical vernacular for your discipline; I am also curious if this term is a specific measurement of “fit”.
Response: Thank you for your suggestion. The term "excellent" was indeed an inaccurate description, and we have adopted your recommendation to use "strong" instead. In psychological research, model fit (CFI = 0.912, RMSEA = 0.068) is an important indicator of the structural validity of measurement tools—both CFI > 0.9 and RMSEA < 0.08 meet the conventional standards for model fit in social science research.
(17)Line 410: This level of heading does not require a number; instead, the title can be bolded to follow APA format. Otherwise, numbering is helpful for organization and clarity.
Response: Thank you for your suggestion. The author has removed the number from this level of heading and bolded the font to comply with APA formatting standards.
(18)Lines 411 & 412: This content warrants source citation(s).
Response: Thank you for your reminder. The author has supplemented references for the normality test and common method bias test.
(19)Line 426: This content warrants source citation(s).
Response: Thank you for your reminder. The author has supplemented references for the partial correlation analysis and Hayes' PROCESS macro.
(20)Line 429: End quotation is missing.
Response: Thank you for your reminder. The author has added the closing quotation mark.
Results
(1)The results section seems to succinctly convey key findings that answer the research questions proposed. Overall, I think this section is strong. One recommendation to strengthen this section would be to elaborate on several of the key findings across variables such as gender, age, years of service, marital status, professional title, and student caseload. Although these are all provided in the table, explicit details provided in the narrative can highlight patterns and nuanced findings for your audience.
Response: Thank you for your valuable suggestion. We fully understand the merit of your idea to elaborate on the key findings related to demographic variables (such as gender, age, and years of service) in the results section—highlighting data patterns through narrative can indeed help readers more intuitively grasp subtle characteristics.
However, after careful consideration, we ultimately decided not to elaborate on these contents in the results section. The main reason is that the core goal of this study focuses on the mechanism relationships between professional identity, self-control, and occupational deviance. Demographic variables are only included as basic characteristics in descriptive statistics (the complete distribution has been presented in Table 1) and are not incorporated into core hypothesis testing or mechanism analysis. If we additionally elaborate on these details, it may distract attention from the core findings of the study. Moreover, considering the limitation of the article length, we hope to prioritize retaining the in-depth interpretation of the relationships between core variables and the results of hypothesis verification.
This trade-off is indeed regrettable, but it is made to keep the discussion more closely centered on the research focus. If there is an opportunity for extended analysis in the future, we will consider supplementing relevant content. Thank you again for your reminder, which has helped us further clarify the priority of information presentation.
(2)Line 456: This content warrants source citation(s); where does the acceptable standard derive from?
Response: Thank you for your reminder. Regarding the acceptable standards (such as χ²/df, GFI, CFI, RMSEA, etc.), they are mainly based on the following authoritative literature:
Hu, Li‐tze, & Bentler, P. M. (1999). Cutoff criteria for fit indexes in covariance structure analysis: conventional criteria versus new alternatives. Structural Equation Modeling, 6(1), 1-55. https://doi.org/10.1080/10705519909540118
(3)The descriptive analysis of the key variables was well organized in providing the specific results, reminding the reader of the hypothesis and how this relates to the results, and ending each paragraph with a digestible synopsis of how the findings can be interpreted in lay terms
Response: Thank you for your recognition of the descriptive analysis section. We will continue to maintain this clear and interconnected presentation style.
(4)Line 491: Unless I am missing something, it is not evident how an absolute value exceeding 0.8 is relevant to the findings provided in Table 2. I understand that this may have been included for methodological rigor, but I wanted to confirm that this was relevant to retain.
Response: Thank you for your question. This paragraph aims to determine whether there is multicollinearity among variables. The study verified this through two methods: first, referring to Wu’s (2010) standard to check if the absolute value of the correlation coefficient between variables exceeds 0.8; second, combining VIF values for further judgment. The mention that "correlation coefficients do not exceed 0.8" is intended to preliminarily indicate that there is no significant multicollinearity among variables, which is directly related to the methodological rigor of the study.
(5)Lines 493 & 494: This content warrants source citation(s).
Response: Thank you for your careful attention. References have been supplemented for this part of the content.
(6)Line 502 (i.e., Table 2): Provide a label in the first column, and capitalize “occupational” under such column to be consistent with other items.
Response: Thank you for your careful reminder. A label has been added to the first column of Table 2, and the first letter of "occupational" in this column has been capitalized to maintain consistency with the format of other items.
(7)Line 522 (i.e., Table 3): the SE columns for Models 2 and 3 do not have enough room for the number of characters, resulting in a spacing error that diminishes clarity of the table; I recommend adjusting both columns to accommodate all necessary characters.
Response: Thank you for your careful observation. The font size in Table 3 has been adjusted to ensure the content is fully displayed.
(8)Line 522 (i.e., Table 3): Content across the table seems jumbled; the numbers run together across columns, diminishing clarity. I recommend breaking each of the three models into their own tables, or reorganizing/reformatting the table.
Response: Thank you for your suggestion. The font size and layout of Table 3 have been re-adjusted to make the table content more neat and clear, improving readability.
Discussion
(1)The content in this section is logically structured to address how the current study relates to previous research. Adding a brief conclusion paragraph that transitions to the summary would enhance clarity and flow.
Response: Thank you for your suggestion. The author has added a brief concluding statement at the end of the discussion section (revised manuscript lines 663-669) to better transition to the summary.
(2)Line 536: I recommend utilizing more tentative language; specifically, “due to” can be interpreted as absolute when there are other potential extraneous factors or explanations.
Response: Thank you for your reminder; your comment is very much to the point. Using "due to" could indeed come across as absolute, so the author has replaced it with "may" to make the expression more tentative, in line with the rigor of academic discussion.
(3)Line 548: Remove the comma after “Song, 2024)”
Response: Thank you for your attention. After rechecking, there is indeed no comma after (Liu & Song, 2024). The misunderstanding may have arisen from a formatting display issue, and we would like to clarify this.
(4)Line 551: The beginning sentence of this paragraph seems disjointed from the following sentence, but are then later explained together, starting on line 554; I recommend reformatting this paragraph for flow and clarity.
Response: Thank you for your suggestion. The entire content of this paragraph has been optimized, with improved sentence transitions and logical structure. The revised version is more coherent and fluent than the original.
(5)Lines 576-578: This content warrants source citation(s). Lines 583-584: This content warrants source citation(s). Lines 593-600: This content warrants source citation(s). Lines 608-609: This content warrants source citation(s). Lines 617-619: This content warrants source citation(s).
Response: Thank you for your suggestion. Relevant references have been supplemented for this content.
(6)Line 591: Include the page number within the in-text citation for quoted text.
Response: Thank you for your attention. After checking, there is indeed no quoted text in line 591, and we would like to clarify this.
(7)Lines 594-595: The ending quotation marks are missing. Additionally, I recommend expanding upon the utilization of arrows and providing further explanation on the directionality of the dimensions.
Response: Thank you for your suggestion. Considering that explaining the arrows might be overly lengthy and deviate from the core argument, the author has removed the arrow symbols and simplified the related expressions to keep the content focused on key points.
(8)Line 623: To increase flow, I recommend combining both sentences with a comma and dropping “this support” to reflect “...frameworks, which can...”, or something to this effect to mitigate redundancy.
Response: Thank you for your suggestion. This part has been revised in accordance with your comments to enhance fluency and reduce redundancy.
Summary
(1)The summary is presented thoroughly and thoughtfully, organizing the key findings and limitations in a logical manner. Adding a brief conclusion paragraph would enhance the organization and flow of the manuscript.
Response: Thank you for your suggestion. A concluding paragraph has been added at the end of the summary section to improve the organization and flow of the manuscript.
(2)Lines 655-658: This content warrants source citation(s). Lines 661-666: This content warrants source citation(s). Lines 672-675: This content warrants source citation(s). Lines 678-680: This content warrants source citation(s).
Response: Thank you for your suggestion. Relevant references have been supplemented for this content.
(3)Line 669: Beginning quotation marks are missing.
Response: Thank you for your careful reminder. The opening quotation mark has been added.
(4)Line 690: This content warrants source citation(s). A page number needs to be included for in-text citations for quoted text. If the quoted text here is to indicate a term the current research team has coined, make this detail explicit in the narrative.
Response: Thank you for your suggestion. A reference has been supplemented for this content. The content in the parentheses is merely explanatory text, not a technical term, and we would like to clarify this.
References
(1)Italicize the volume number (i.e., following the article/book title) for all article sources to be consistent with APA format.
Response: Thank you for your suggestion. The volume numbers of all article sources have been italicized as required to comply with APA formatting standards.
(2)Line 836: This citation is out of order; Muraven needs to follow Moore.
Response: Thank you for your reminder. The order of these two citations has been adjusted.
(3)Line 860: Is this Wang, J. P., similar to the previous citation author on line 858? If so, make sure they are consistent. If not, the Wang, J. citation would go before Wang, J. P. citation.
Response: Thank you for your reminder. These refer to two different authors, and their order has been adjusted.

Reviewer 2 Report
Comments and Suggestions for Authors
This is important research because university counselors in China play a crucial role that, in a broader context, can impact the country’s future labor market.
detailed feedback:
1. The role of university counselors varies across countries and educational systems. In particular, the role in China differs significantly from that in Western countries. These differences should be reflected in the practical implications, research limitations, and objectives of the study. In this case, only Chinese universities can serve as concrete reference points.
2. To further emphasize the importance of this research, the potential consequences of misconduct by university counselors could be outlined – for example, damage to the reputation of universities and a reduction in the quality of human capital entering the labor market.
3. In the chapter "Study Design – Inclusion Criteria," it is stated that one year of professional experience was required for participation in the study. However, it is questionable whether this is sufficient to establish stable behavioral patterns, especially considering that the onboarding phase alone can take up to six months.
Author Response
- The role of university counselors varies across countries and educational systems. In particular, the role in China differs significantly from that in Western countries. These differences should be reflected in the practical implications, research limitations, and objectives of the study. In this case, only Chinese universities can serve as concrete reference points.
Response: Thank you for your insightful comments. It is indeed true that the role of university counselors varies significantly across national education systems, with the responsibilities and positioning of university counselors in China featuring distinct local characteristics compared to those in Western countries. We have taken note of this difference in our research and specifically addressed it in the sections on practical implications, limitations, and research objectives. We clearly define Chinese universities as the specific research context and reference framework to ensure the relevance and applicability of our findings.
- To further emphasize the importance of this research, the potential consequences of misconduct by university counselors could be outlined – for example, damage to the reputation of universities and a reduction in the quality of human capital entering the labor market.
Response: Thank you for your valuable suggestion. The potential consequences you mentioned—such as damage to a university's reputation and a decline in the quality of human capital entering the labor market—are highly consistent with the core impacts we focus on in our research. In fact, this is already addressed in lines 144-146 of the article: "Such behaviors not only undermine the credibility of educational institutions but also exert a significant negative influence on students' value systems through social modeling effects." The shaping of students' value systems is closely linked to the quality of human capital. Going forward, we will consider further strengthening the articulation of this connection to more clearly highlight the practical significance of the research.
- In the chapter "Study Design – Inclusion Criteria," it is stated that one year of professional experience was required for participation in the study. However, it is questionable whether this is sufficient to establish stable behavioral patterns, especially considering that the onboarding phase alone can take up to six months.
Response: Thank you for your careful attention and professional inquiry. The inclusion criterion of "one year of professional experience" is based on the actual career development context of university counselors in China: New recruits must undergo a one-year probationary period, during which they have not yet obtained formal teacher qualification certificates or position appointments, and have not fully entered a stable professional role. Additionally, there is a high turnover rate during this period, and professional behavioral patterns remain undetermined. Therefore, selecting counselors with at least one year of experience as research subjects aims to ensure that their professional behaviors are relatively stable and representative, thereby enhancing the reliability of the research data. This consideration is also consistent with the common practices in the management of university counselor teams in China.

Reviewer 3 Report
Comments and Suggestions for Authors
Thank you for asking me to review this manuscript — it has also helped me to sharpen my own thinking. Aiming to “explore the negative predictive influence of professional identity on occupational deviance among university counselors, as well as to assess the moderating effect of self-control on this relationship,” the article provides, in my opinion, a largely coherent and scientifically sound contribution with implications for both educational practice and theory.
Nonetheless, I believe that the manuscript would benefit from slight revisions in a few places in order not to unintentionally confuse readers. From the perspective of argumentative coherence, it is initially somewhat surprising that the sentence “critical studies addressing instances of occupational deviance remain notably scarce” (lines 62–63) appears rather abruptly. I would have appreciated a clearer lead-in to the focus on occupational deviance. Furthermore, the statement “the generative mechanisms underlying occupational deviance exhibit dual pathological characteristics” (lines 68–69) introduces a specific narrative of occupational deviance that does not fully align with my own understanding of the term. In my reading, the authors seem to suggest that occupational deviance is inherently deficit-oriented, which does not necessarily have to be the case. Deviance can indeed serve as a trigger for creative or otherwise progressive processes, especially when existing institutionalized rules conflict with the value system of a profession — in which case institutional blind spots may even be advantageous. To cut a long story short: in my view, a definition of occupational deviance should be provided prior to lines 68–78 so as not to unsettle readers who, like me, understand “deviance” as a more neutral term.
In addition, Section 1.3 describes self-control as a “personality trait” (line 202), only to then characterise it in line 203 as “an individual’s capacity.” I would suggest replacing the former with “an individually acquired disposition,” so as to avoid confusion as to whether self-control is seen as learnable — which, according to my reading, the authors indeed suggest (e.g., lines 290–291). I consider the statement “Robust external constraints are vital for maintaining high standards in educational institutions” (lines 538–539), and those following it, to be somewhat problematic, as it is programmatic and overstated. The relationship between high standards in educational institutions and robust external constraints was not examined in this study; more cautiously, one might suggest that such constraints could be a reason for the low prevalence of occupational deviance among counselors in Chinese higher education.
The research design — described in sufficient detail to ensure reproducibility — appears well suited to answering the research questions posed. The figures, tables, and interpretations are also accessible, accurate, and consistent. However, I wonder why the scale is named “professional volition.” Does this still capture the theoretically introduced dimension of “role commitment”? If possible, a clarifying footnote would be helpful.
The referencing mostly includes recent and relevant publications where conceptually appropriate.
Comments on the Quality of English LanguageFormally, I noted the following issues:
• Number “1” circled in the abstract;
• Unnecessary space in line 46;
• Unnecessary quotation marks in lines 65, 466, and 594;
• Superscript comma and unnecessary space in line 123;
• Unclear sentence in lines 141–143;
• Missing spaces in lines 148 and 506;
• “Schemas” (line 174) should read “schemes”;
• Missing verb (e.g., “predict”) in line 294;
• Missing quotation marks in lines 317 and 429;
• Unnecessary comma and unnecessary period in line 429;
• In my opinion, Table 3 should also be rotated to landscape orientation in order to resolve the somewhat awkward formatting.
Author Response
Revision Explanation for Reviewer 3(All revisions have been highlighted)
- Thank you for asking me to review this manuscript — it has also helped me to sharpen my own thinking. Aiming to “explore the negative predictive influence of professional identity on occupational deviance among university counselors, as well as to assess the moderating effect of self-control on this relationship,” the article provides, in my opinion, a largely coherent and scientifically sound contribution with implications for both educational practice and theory.
Response: Thank you for your recognition of this study and your valuable review comments. Your feedback has not only helped improve the manuscript but also provided us with new perspectives for reflection. We fully agree with your assessment of the core contributions of the study and its theoretical and practical value. Going forward, we will continue to refine the details to further enhance the coherence and scientific rigor of the research.
- Nonetheless, I believe that the manuscript would benefit from slight revisions in a few places in order not to unintentionally confuse readers. From the perspective of argumentative coherence, it is initially somewhat surprising that the sentence “critical studies addressing instances of occupational deviance remain notably scarce” (lines 62–63) appears rather abruptly. I would have appreciated a clearer lead-in to the focus on occupational deviance. Furthermore, the statement “the generative mechanisms underlying occupational deviance exhibit dual pathological characteristics” (lines 68–69) introduces a specific narrative of occupational deviance that does not fully align with my own understanding of the term. In my reading, the authors seem to suggest that occupational deviance is inherently deficit-oriented, which does not necessarily have to be the case. Deviance can indeed serve as a trigger for creative or otherwise progressive processes, especially when existing institutionalized rules conflict with the value system of a profession — in which case institutional blind spots may even be advantageous. To cut a long story short: in my view, a definition of occupational deviance should be provided prior to lines 68–78 so as not to unsettle readers who, like me, understand “deviance” as a more neutral term.
Response: Thank you for your constructive suggestions. Regarding the clarification of the concept of occupational deviance, we have added a specific explanation of university counselors' occupational deviance in lines 78–79, defining it explicitly as "behaviors that violate professional ethical norms and role expectations (such as bias in handling student affairs or neglect of educational responsibilities)." By delineating the research scope through concrete examples, we have avoided a generalized understanding of the term "deviance" and distinguished it from the "creative deviance" you mentioned, ensuring the consistency and specificity of the term's usage. We will further check the logical flow of the text to strengthen the fluency of the argument.
- In addition, Section 1.3 describes self-control as a “personality trait” (line 202), only to then characterise it in line 203 as “an individual’s capacity.” I would suggest replacing the former with “an individually acquired disposition,” so as to avoid confusion as to whether self-control is seen as learnable — which, according to my reading, the authors indeed suggest (e.g., lines 290–291). I consider the statement “Robust external constraints are vital for maintaining high standards in educational institutions” (lines 538–539), and those following it, to be somewhat problematic, as it is programmatic and overstated. The relationship between high standards in educational institutions and robust external constraints was not examined in this study; more cautiously, one might suggest that such constraints could be a reason for the low prevalence of occupational deviance among counselors in Chinese higher education.
Response: Thank you for your professional analysis and meticulous corrections. We fully endorse your views and have revised the description of self-control to: "the ability of individuals to autonomously regulate their thoughts, emotions, and behaviors to achieve predetermined goals or adhere to specific standards, exhibiting certain trait-like characteristics while possessing significant acquired plasticity, involving behavioral patterns such as inhibiting impulses, overcoming short-term temptations, and following long-term goal-oriented tendencies." This formulation not only reflects the stable characteristics of self-control but also emphasizes its learnable and trainable nature, aligning with the practical suggestion in the study that "self-control can be enhanced through training." Additionally, regarding the statement about external constraints, we have revised it to "external constraints may be one of the reasons for the low incidence of occupational deviance among university counselors in China," making it more in line with the scope of the study and avoiding overgeneralization.
- The research design — described in sufficient detail to ensure reproducibility — appears well suited to answering the research questions posed. The figures, tables, and interpretations are also accessible, accurate, and consistent. However, I wonder why the scale is named “professional volition.” Does this still capture the theoretically introduced dimension of “role commitment”? If possible, a clarifying footnote would be helpful.
Response: Thank you for your careful attention. Regarding the naming logic of the "professional will" dimension and its relationship with "role commitment," we have added an explanatory footnote: When developing the scale, Liu Shiyong (2018) defined "professional will" as "an individual's initiative and resilience in persisting in career choices and fulfilling professional responsibilities when facing occupational stress, burnout, or external temptations," focusing on "sustained motivation to overcome difficulties," taking into account the uniqueness of counselors' careers, which involve coping with multiple pressures and role ambiguity. In contrast, "role commitment" emphasizes emotional attachment and normative identification with one's professional role. Although both are core elements of professional identity, the former highlights proactive persistence in stressful situations, while the latter focuses on a sense of belonging to the role itself. This clarification aims to specify the connections and differences between the two concepts, enhancing the clarity of their usage.
- The referencing mostly includes recent and relevant publications where conceptually appropriate.
Response: Thank you for your attention to the citation of literature. In elaborating on concepts and providing theoretical support, this study has prioritized the use of relevant representative studies from the past five years as well as classic literature, striving to balance cutting-edge relevance and authority. We will systematically review the references to ensure that citations for key concepts and core viewpoints are not only relevant to the research topic but also reflect the latest developments in the field.
- Comments on the Quality of English Language
Formally, I noted the following issues:
- Number “1” circled in the abstract;
- Unnecessary space in line 46;
- Unnecessary quotation marks in lines 65, 466, and 594;
- Superscript comma and unnecessary space in line 123;
- Unclear sentence in lines 141–143;
- Missing spaces in lines 148 and 506;
- “Schemas” (line 174) should read “schemes”;
- Missing verb (e.g., “predict”) in line 294;
- Missing quotation marks in lines 317 and 429;
- Unnecessary comma and unnecessary period in line 429;
- In my opinion, Table 3 should also be rotated to landscape orientation in order to resolve the
Response: Thank you for pointing out the specific formatting and expression issues. Regarding all the details you mentioned—including the notation of the number "1" in the abstract, redundant spaces, improper use of quotation marks, errors in superscript symbols and spaces, unclear sentence expressions, missing spaces, adjustment of terminology, missing verbs, misuse of punctuation marks, and the formatting optimization of Table 3—the authors have checked each item and made corrections to ensure the text is formatted standardly and expressed accurately and clearly. Thank you for your meticulous review; these adjustments have further enhanced the rigor of the manuscript.
